# Genomic Surveillance Uncovers a 10-Year Persistence of an OXA-24/40 *Acinetobacter baumannii* Clone in a Tertiary Hospital in Northern Spain

**DOI:** 10.3390/ijms25042333

**Published:** 2024-02-16

**Authors:** Maitane Aranzamendi, Kyriaki Xanthopoulou, Sandra Sánchez-Urtaza, Tessa Burgwinkel, Rocío Arazo del Pino, Kai Lucaßen, M. Pérez-Vázquez, Jesús Oteo-Iglesias, Mercedes Sota, Jose María Marimón, Harald Seifert, Paul G. Higgins, Lucía Gallego

**Affiliations:** 1Respiratory Infection and Antimicrobial Resistance Group, Microbiology Department, Infectious Diseases Area, Biogipuzkoa Health Research Institute, Osakidetza Basque Health Service, Donostialdea Integrated Health Organization, 20014 San Sebastián, Spain; maitane.aranzamendizaldumbide@osakidetza.eus (M.A.); josemaria.marimonortizdez@osakidetza.eus (J.M.M.); 2*Acinetobacter baumannii* Research Group, Department of Immunology, Microbiology and Parasitology, Faculty of Medicine and Nursing, University of the Basque Country UPV/EHU, 48940 Leioa, Spain; sandrasanchezurtaza@gmail.com; 3Institute for Medical Microbiology, Immunology and Hygiene, Faculty of Medicine and University Hospital Cologne, University of Cologne, 50935 Cologne, Germany; kyriaki.xanthopoulou@uk-koeln.de (K.X.); tessa.burgwinkel@uk-koeln.de (T.B.); rocio.arazo-del-pino@uk-koeln.de (R.A.d.P.); kai.lucassen@uk-koeln.de (K.L.); harald.seifert@uni-koeln.de (H.S.); 4German Center for Infection Research (DZIF), Partner Site Bonn-Cologne, 50935 Cologne, Germany; 5National Center of Microbiology, Reference and Research Laboratory for Antibiotic Resistance, ISCIII, Centro de Investigación Biomédica en Red, Enfermedades Infecciosas (CIBERINFEC), 28220 Madrid, Spain; mperezv@isciii.es (M.P.-V.); jesus.oteo@isciii.es (J.O.-I.); 6Clinical Laboratory Management Department, IIS Biodonostia Health Research Institute, University Hospital Donostia, 20014 Donostia, Spain; mercedes.sotabusselo@osakidetza.eus; 7Institute of Translational Research, CECAD Cluster of Excellence, University of Cologne, 50935, Cologne, Germany

**Keywords:** *Acinetobacter baumannii*, carbapenem resistance, hospital outbreaks, sequencing, plasmid, persistence

## Abstract

Infections caused by carbapenem-resistant *Acinetobacter baumannii* are a global threat causing a high number of fatal infections. This microorganism can also easily acquire antibiotic resistance determinants, making the treatment of infections a big challenge, and has the ability to persist in the hospital environment under a wide range of conditions. The objective of this work was to study the molecular epidemiology and genetic characteristics of two *bla*_OXA24/40_ *Acinetobacter baumannii* outbreaks (2009 and 2020-21) at a tertiary hospital in Northern Spain. Thirty-six isolates were investigated and genotypically screened by Whole Genome Sequencing to analyse the resistome and virulome. Isolates were resistant to carbapenems, aminoglycosides and fluoroquinolones. Multi-Locus Sequence Typing analysis identified that Outbreak 1 was mainly produced by isolates belonging to ST3^Pas^/ST106^Oxf^ (IC3) containing *bla*_OXA24/40_, *bla*_OXA71_ and *bla*_ADC119_. Outbreak 2 isolates were exclusively ST2^Pas^/ST801^Oxf^ (IC2) *bla*_OXA24/40_, *bla*_OXA66_ and *bla*_ADC30_, the same genotype seen in two isolates from 2009. Virulome analysis showed that IC2 isolates contained genes for capsular polysaccharide KL32 and lipooligosacharide OCL5. A 8.9 Kb plasmid encoding the *bla*_OXA24/40_ gene was common in all isolates. The persistance over time of a virulent IC2 clone highlights the need of active surveillance to control its spread.

## 1. Introduction

*Acinetobacter baumannii* was listed in 2008 as part of the broad ESKAPE group (*Enterococcus faecium*, *Staphylococcus aureus*, *Klebsiella pneumoniae*, *A. baumannii*, Pseudomonas aeruginosa, and Enterobacter species) [1,2] and it has been recognized as such a big threat, that the World Health Organization (WHO) in 2017 classified carbapenem-resistant *A. baumannii* (CRAb) as one of the “Priority 1: Critical group” organisms for which new antibiotics are urgently needed [3,4].

*A. baumannii* is an opportunistic pathogen frequently responsible for outbreaks in health-care facilities, particularly in Intensive Care Units (ICUs) [5]. A wide range of virulence factors are responsible for pathogenesis and high mortality of *A. baumannii*, contributing to pathogen survival in stressful conditions [6,7,8,9]. *A. baumannii* can also easily acquire antibiotic resistance determinants rendering multiple antimicrobials useless, hindering patient treatment [10,11]. To add to this, the organism’s ability to survive under a wide range of environmental conditions and to persist for extended periods of time on dry surfaces make it a frequent endemic nosocomial pathogen [12,13,14,15].

Moreover, the persistence of *A. baumannii* in the hospital environment could potentially be favoured by reduced susceptibility to commonly used biocides and this seems to induce co-resistance to some antimicrobial agents (carbapenems, aminoglycosides, quinolones and tetracyclines). These two factors, persistence and multidrug resistance (MDR), have been clearly associated with the epidemic behaviour of some successful clones of *A. baumannii* [16].

High resistance rates to last resort antibiotics, such as carbapenems, are of major concern [17]. OXA enzymes are the most prevalent carbapenem-resistance determinant in *A. baumannii* with up to six families described i.e., the intrinsic OXA-51 and the acquired OXA-23, OXA-24/40, OXA-58, OXA-143 and OXA-235 [18]. The OXA-24/40 carbapenemase was first identified in a multidrug-resistant isolate from a hospital in Northern Spain [19] but quickly was confirmed as an endemic enzyme in the Iberian peninsula [20]. Although first described as chromosomally encoded, it is also harboured by plasmids [21].

International Clone 2 (IC2) is the most prevalent clone in most countries of Southern Europe and in the United States, whereas other less prevalent clones have been described in Central and South America (IC5 and IC7) and in Africa and the Middle East (IC9) [22,23,24,25,26]. In Spain IC2 isolates are the predominant genotype and have been associated with increased antimicrobial resistance (AMR), especially to carbapenems due to the presence of OXA-type carbapenemases, mainly OXA-23 and less frequently OXA-24/40 [27,28,29].

Although significant advances have been made in our understanding of *A. baumannii* over the years, many unanswered questions about the spread and evolution of carbapenem-resistant isolates remain unclear [30]. The aim of the present study was to investigate the molecular epidemiology and genetic features of two OXA-24/40-producing CRAb outbreaks at a large tertiary hospital in Northern Spain in 2009 and 2019–2021, respectively.

## 2. Results

### 2.1. Species Identification and Antimicrobial Susceptibility

All isolates were confirmed as *A. baumannii* according to the MALDI-TOF, *gyr*B multiplex PCR, and identification of the intrinsic *bla*_OXA-51_. Antimicrobial susceptibility testing showed values above the EUCAST breakpoints (Version 13.1, 2023) for all β-lactams (including imipenem and meropenem, MICs of >8 mg/L and of >4 mg/L respectively), aminoglycosides (MICs of >4 mg/L against gentamicin and tobramycin except for HUC-106 and HUC-109 with MICs of 4 mg/L; MICs of >8 mg/L against amikacin for the 86.8% of the isolates), fluoroquinolones (MICs of >1 mg/L against ciprofloxacin and levofloxacin), sulphonamides (MICs of >4/76 mg/L against cotrimoxazole except for HUC-106 and HUC-109 with MICs of 2/38 mg/L), while all the isolates except for HUC-88 were susceptible to colistin (MIC ≤ 2 mg/L) (Appendix A).

### 2.2. MLST/cgMLST Analysis

The isolates were assigned to the Pasteur sequence type (ST) 2 (*n* = 24), ST3 (*n* = 12) and ST49 (*n* = 2) and following the Oxford scheme isolates were clustered in ST801 (*n* = 23), ST350 (*n* = 1), ST106 (*n* = 12) and ST128 (*n* = 2).

Using cgMLST analysis the isolates clustered with previously identified international clones (IC). In detail the isolates recovered in 2009 were identified as IC3 (*n* = 11), IC2 (*n* = 2) and unique (*n* = 2). The IC3 isolates formed a cluster of closely related isolates differing in up to 8 alleles. Interestingly, the control isolate HUC-A11 (*n* = 1) recovered in 2002 clustered also together with the IC3 collected in 2009, suggesting that the latter isolates might have been involved in inter-hospital transmission.

Using cgMLST all isolates recovered between 2020–2021 and samples collected from potentially contaminated objects from 5 different patient rooms, belonged to IC2 and formed a cluster of closely related isolates differing in up to 2 alleles. It is worth mentioning that the two IC2 isolates collected in 2009 clustered together with the recovered between 2020–2021 and were considered as closely related. Concerning the control strains, we found that on one hand, although HUC-SM28 also belonged to IC2 it was 395 alleles different from the IC2 isolates from the outbreaks, so that, was considered unrelated. On the other hand, strain HUC-A11 isolated in 2002, which was also identified as IC3, clustered with IC3 isolates from Outbreak 1 (Figure 1, in cyan).

### 2.3. Resistome 

Different variants of the *bla*_OXA-51_ gene were found; *bla*_OXA-66_ (IC2), *bla*_OXA-71_ (IC3) and *bla*_OXA-98_. All isolates encoded the carbapenemase *bla*_OXA-24/40_ gene, and a novel *bla*_OXA-24/40-like_ variant, named *bla*_OXA-1040_ (GenBank Accession Number OK271078), identified for the first time in isolate HUC-A11 (Table 1). One isolate (HUC-101) also harboured a *bla*_OXA-58_ gene.

Two variants of intrinsic *bla*_ADC-51_ (*bla*_ADC-30_ and *bla*_ADC-119_) were predomminantly detected in the isolates, with the variants reflecting the cgMLST cluster pattern, i.e., *bla*_ADC-30_ was found in all IC2 isolates, while *bla*_ADC-119_ was associated with IC3. Furthermore, the intrinsic *bla*_ADC-1_ and *bla*_ADC-241_ variants were associated with clonal lineages IC2 and NA respectively, as confirmed by Oxford and Pasteur MLST, as well as cgMLST.

Furthermore, the IC3 isolates encoded the *aad*B-*like*, *aph*(*3*′)-*Via* and *sul1*, while all the IC2 isolates encoded the *aac*(*3*)-Ia-like, *aad*A1, *aph*(*3*′)-*VIa*-like, *str*A *and str*B, the sulphonamide resistance gene *sul2* in addition to *sul1*, as well as the tetracycline resistance determinant *tet*(B)-like. All fluoroquinolone resistant isolates showed a double substitution Ser81-Leu and Ser84-Leu in *Gyr*A and *Par*C, respectively (Table 1).

### 2.4. Virulome

Results are summarised in Table 2. We found genes coding for the *Acinetobacter* trimeric autotransporter (*ata*); host cell adhesion (*pil* and *fim*); the two-component signal transduction system, biofilm controlling response regulator and sensor kinase (*bfm*RS); efflux pump operon (*ade*FGH); the quorum-sensing system (*aba*IR), consisting of the acyl-homoserine-lactone synthase AbaI and the DNA-binding HTH-domain-containing protein AbaR; the chaperonusher pilus CsuC/D/E (*csu*CDE); the biofilm-associated protein Bap (*bap*) as well as the Bap-like proteins Blp1 and Blp2 (bpl1, bpl2); the outer membrane protein A (*ompA*); the polysaccharide poly-β-(1,6)-N-acetyl glucosamine (PNAG) encoded by gene cluster *pgaABCD*; and the coagulation targeting metallo-endopeptidase of *A. baumannii* (*cpa*A).

To mention, the chaperonusher pilus CSU-encoding genes were only detected in IC2, while capsular gene *cpa*A was only present in isolates HUC-106 and HUC-109. Although most of them were homogeneously present or not in all the isolates of each clone, *ata* gene was detected in the 78% of IC2 isolates, *bfm*S in the 82% of IC3 isolates and *bap* gene was present in the 36% of isolates of IC3. The K32, K1 and K11 capsular types, and OCL5, OCL1 and OCL8 outer core loci were found in our IC2, IC3 and NA strains, respectively. 

The most important virulence genes in *A. baumannii* related to adherence, biofilm formation, exotoxins, exoenzymes, iron uptake, capsular polysaccharide and lipooligosaccharide outer core loci are summarized in Table 2.

### 2.5. OXA-24/40 Encoding Plasmids

Using a hybrid assembly approach, it was identified that the *bla*_OXA-24/40_ was encoded on a plasmid of 8.9 Kb in size in the isolates HUC-90, HUC-99 and Aba-1527 (Figure 2). The OXA-24/40-encoding plasmid was identical in the isolates belonging to IC2 clone (HUC-99 and Aba-1527) with a 100% coverage and identity, and highly similar to the plasmid of the isolate belonging to IC3 (HUC-90) with a 100% coverage and 99% identity.

Plasmid annotation revealed ORFs encoding for the carbapenemase OXA-20/40; a RepM replicon (Rep-3 superfamily, GR2); a toxin-antitoxin system (BrnT-BrnA) that is involved in vertical stability; TonB-dependent receptor, related to the transmission of signals from the outside of the cell leading to transcriptional activation of target genes; DIP1984 family protein encoding a hypothetical protein as well as three unidentified hypothetical proteins.

### 2.6. Transfer of bla_OXA-24/40_ by Electroporation and Conjugation Experiments

The three *bla*_OXA24/40_-encoding plasmids (pHUC-90, pHUC-99 and pAba-1527) were transferable by electroporation into *A. baumannii* ATCC17978 and the presence of *bla*_OXA24/40_ gene in the obtained transformants was confirmed by PCR. Furthermore, conjugation experiments showed that the OXA-24/40-encoding plasmids present in the IC2 isolates, HUC-99 and HUC-86/Aba1527, were transferred into the recipient *A. baumannii* BM4547. Conjugation experiments with the control strains were not successful.

## 3. Discussion

The relevance of carbapenem-resistant *A. baumannii* as a healthcare associated pathogen (HAP) is in part due to its ability to survive for long periods in hospital environment; therefore, it should be taken into account as a relevant source of *A. baumannii* outbreaks [13,31].

The present study describes two independent outbreaks of isolates harbouring the *bla*_OXA-24/40_ gene caused by two different clones in 2009 and 2020, belonging to IC3 and IC2, respectively.

Isolates showed a multidrug resistance phenotype, and besides phenotypical resistance to β-lactams, including carbapenems, most of the isolates of our collection were also resistant to sulphonamides, aminoglycosides and fluroquinolones, restricting the therapeutic options to colistin (except for isolate HUC-88, which was resistant to it), minocycline and tigecycline. Unfortunately, at the time of the outbreaks, susceptibility testing to novel compounds was not available in our hospital. Especially worrying is that most of the isolates in this study were obtained from the ICU section of the hospital. 

Clonal relatedness analysis showed that the majority of the isolates of outbreak 1 belonged to IC3, more precisely they were assigned to ST3 (Pasteur) and ST106 (Oxford). This clone was considered a major clone in the past, but nowadays it has been replaced by IC2 and it is reported sporadically in Spain, United States of America and South Africa [32]. In contrast, isolates from outbreak 2 predominantly belonged to IC2, being ST2 (Pasteur) the most prevalent lineage, which is the most reported ST among CRAB isolates in the Mediterranean countries [33].

Outbreak 2 was caused by isolates belonging to IC2 and were identical to the IC2 isolates from outbreak 1, showing the persistence and stability of this strain over an eleven-year period. The homology of these isolates with the control strains showed coincidence between IC3 isolates from outbreak 1 with HUC-A11, isolated in 2002, but a wide distance (more than 300 alleles different) between IC2 isolates and HUC-SM28 recovered in 1999. It should also be noted that HUC-SM28 was the first *A. baumanii* isolate in which the *bla*_OXA-24/40_ carbapenemase variant was described (AF509241.1) [19,34]. In this study, sequencing of the *bla*_OXA-24/40_ gene in HUC-A11 isolate allowed us to identify a new variant of this gene, the *bla*_OXA-1040_ (OK271078).

Nowadays the genotype IC2 is the most commonly isolated clone in Mediterranean countries, especially in the ICU where it also has a higher rate of carbapenem resistance compared to the other *A. baumannii* international clones [35]. In the present study, isolates belonging to IC2 not only persisted along time, but also posessed multiple resistance determinants such as *bla*_OXA-24/40_, as well as resistance against aminoglycosides (*aac*(*3*)-Ia-*like*, *aad*A1,*aph*(*3′*)-*VIa-like*, *str*A and *str*B), sulphonamides (*sul2* in addition to *sul1*), and tetracycline (*tet*(B)-*like*). Differences between IC2 and IC3 isolates relied on the absence of *tet*(B)-*like*, *sul2*, *stra*A and *stra*B, *aac*(*3*)-Ia, and the presence of *aad*B-*like* in isolates belonging to IC3 as well as the intrinsic OXA and ADC.

The success of *A. baumannii* as a nosocomial pathogen is due to not only antibiotic resistance, but also to virulence factors such as the capsule [36]. Interestingly, IC2 strains were KL32 capsular type that has been described as one of the most virulent loci. Moreover, KL32 has been associated with a significant increase of cell survival when treated with specific phages, and shows another way to favour long persistence, which may explain, among other factors, the long survival of the IC2 isolates of the present study [37,38].

Our results showed that, IC2 isolates in contrast to the other genotypes, harboured *csu*C, *csu*D and *csu*E genes and exhibited a higher percentage of *bap* genes, both *csu* and *bap*, known to be required in the early steps of biofilm formation. They also lack the *cpa*A gene, related to encapsulation, which conversely may even negatively impact bacterial biofilm formation by increasing cell surface hydrophobicity and may sensitize bacterial cells to UV radiation [39,40]. Isolates belonging to IC2 genotype usually harbour more virulence genes associated with higher capacity to produce biofilms on abiotic surfaces used for medical supplies, e.g., urinary catheters or endotracheal tubes what could reinforce the long persistence of our isolates [6,41,42,43,44,45].

Plasmids represent an ideal vehicle for acquiring and transferring resistance genes in *A. baumannii*. The most complex and large ones can carry transposable elements such as integrons or resistance islands (RI), while small plasmids may even lack conjugative or replicating genes, that may reduce the fitness cost of plasmid replication, enabling plasmid stability over long periods of time [46,47,48,49]. Our results showed an identical plasmid harbouring *bla*_OXA-24/40_ in the IC2 strains, HUC-99 and Aba1527, that persisted intact for a decade and crossed also the IC boundaries as it was also detected with minor variations in the IC3 isolate obtained in 2009. The plasmid belonged to GR2 homology group. Small plasmids of similar size have often been described in different clonal lineages, such as IC1 [48]. Furthermore, these plasmids were able to be transferred into the recipient strains used in this study, showing their role as dissemination mechanism of the *bla*_OXA-24/40_ gene.

There is a wide array of possibilities for *A. baumannii* to disseminate among patients and the hospital setting, including equipment, hospital surfaces, direct transmission between patients and/or healthcare workers, among the most relevant ones [13]. From 2009 to 2020, no CRAb isolates were recovered in our hospital, probably due to the additional measurements that were implemented such as an active screening programme on hospital admission, surveillance cultures from both patients and abiotic surfaces in critical hospital units, etc. [50,51,52].

The coronavirus disease 2019 (COVID-19) pandemic posed an additional threat and placed pressure on the increase in AMR worldwide, overwhelming health systems and creating a multifactorial problem. Hospitalization, especially in ICUs, increases the chances of healthcare associated infections (HAIs), mainly associated with the rise in the use of invasive devices, such as mechanical ventilation and vascular catheters. Increased length of stay, human resource challenges, that limited the adherence and effectiveness of standard infection prevention practices also contributed to the increase in HAIs [53,54,55]. Under this situation many opportunistic pathogens, such as *A. baumannii*, were able to survive in the nosocomial environment and re-emerged leading to fatal infections [54,55,56,57] as it could have happened during the pandemic, contributing to Outbreak 2. Unfortunately, we cannot ascertain whether the disruption of any measure could contribute in a larger degree or not to the re-emergence and spread in our hospital [2].

The high prevalence of antibiotic resistance as well as the complexity of intertwined resistance mechanisms in *A. baumannii* clinical isolates, highlights the importance of antimicrobial stewardship and active surveillance in clinics, as it poses a major threat to human health [58].

## 4. Materials and Methods

### 4.1. Bacterial Isolates

The molecular epidemiology of thirty-six OXA-24/40-positive CRAb isolates was investigated (Appendix A) through a retrospective descriptive study. Twenty-nine isolates were recovered from 29 patients hospitalized at the Hospital Universitario de Cruces (Barakaldo, Spain) in 2009 (*n* = 15) and 2019–2021 (*n* = 14). The first isolate was selected, while isolates from sterile sites were chosen over respiratory samples, whenever possible. Additionally, seven samples collected in 2020 from potentially contaminated surfaces and equipment in the ICU of the same hospital were also studied. The environmental samples were processed using moist gauze following a modified method described previously [59]. As control strains we included two OXA-24/40 CRAb isolates obtained in 1999 (HUC-SM28) and in 2002 (HUC-A11) from two patients attended at the same hospital. 

### 4.2. Species Identification, and Antimicrobial Susceptibility Testing

Species identification was performed using Matrix Assisted Laser Desorption/Ionization-Time of Flight Mass Spectrometry (MALDI-TOF MS) (Maldi Biotyper^®^; Bruker Daltonik, Bremen, Germany) and confirmed as *A. baumannii* by the *gyr*B multiplex PCR and identification of the intrinsic *bla*_OXA-51-like_ gene, as previously described [60].

Susceptibility testing was performed using the VITEK^®^2 system with card AST-N089 22237 (bioMérieux, Marcy-l’Étoile, France) or Negative Combo 38 Panel from MicroScan^®^ WalkAway^®^ 96 Plus System (both Beckman Coulter Inc.; Brea, CA, USA). Minimal inhibitory concentrations (MICs) were interpreted using the resistance breakpoints for *Acinetobacter* spp. from the European Committee on Antimicrobial Susceptibility Testing (EUCAST) breakpoints (Version 13.1, 2023; http://www.eucast.org/clinical_breakpoints/, accessed on 29 June 2023). For tigecycline, the EUCAST PK-PD (Non-species related) breakpoint of 0.5 mg/L for Enterobacterales was used.

### 4.3. DNA Extraction, PCRs, Short-Read Sequencing and Assembly 

DNA was extracted using the Dneasy UltraClean Microbial Kit and used for the OXA-multiplex PCR to detect the carbapenemase-encoding genes *bla*_OXA-51-like_, *bla*_OXA−23-like_, *bla*_OXA−58-like_, *bla*_OXA−24/40-like_, *bla*_OXA−143-like_, and *bla*_OXA−235-like_, as previously described [61,62]. Subsequently, sequencing libraries were prepared using NEB Next^®^ Ultra™ II FS DNA Library Prep Kit for Illumina^®^ (#E7805//#6177; New England Biolabs) for a 250 bp paired-end sequencing run on an Illumina^®^ MiSeq sequencer and the obtained reads were assembled de novo using the Velvet assembler integrated in the Ridom SeqSphere+ v.7.0.4 (Ridom GmbH, Münster, Germany).

### 4.4. Molecular Epidemiology and Determination of Antibiotic Resistance and Virulence Genes

The isolates were investigated by applying a validated core genome Multi-Locus Sequence Typing (cgMLST) scheme, using the Ridom SeqSphere+ v. 8.0.1 software (Ridom GmbH) to generate a minimum spanning tree including 2390 target alleles [63].

Sequence types (STs) were assigned using the Oxford and the Pasteur 7-loci multi-locus sequence typing (MLST) schemes from the genome assemblies using the pubMLST website (https://pubmlst.org/organisms/acinetobacter-baumannii, accessed on 5 October 2021).

The software Resfinder 4.4.1 (http://genepi.food.dtu.dk/resfinder, accessed on 2 September 2021) and CARD (https://card.mcmaster.ca/, accessed on 3 September 2021) were used to identify acquired resistance genes and the beta-lactamase database (http://www.bldb.eu/, accessed on 30 August 2021) was used to identify the *bla*_ADC_ variants.

Adherence, biofilm and quorum sensing-related genes search was performed using the PubMLST website, only exact matches were taken into consideration [64]. The Virulence Factor Database (VFDB) and Kaptive were used to identify exotoxins, exoenzymes, iron uptake, capsular polysaccharide and lipooligosaccharide outer core loci [65,66].

### 4.5. MinION Long-Read Sequencing and Assembly

DNA extraction for long-read sequencing was performed using the MagAttract kit (Qiagen Cat. No. 67653) and the Wizard Genomic Purification kit (Promega Cat. No. A2920) for the DNA isolation according to the manufacturer’s instructions. Libraries were prepared using the 1D Ligation Sequencing Kit (SQK-LSK109) in combination with Native Barcoding Kit (EXP-NBD104; Oxford Nanopore Technologies, Oxford, UK) and were loaded onto a R9.4 flow cell (Oxford Nanopore Technologies). The run was performed on a MinION MK1b device. Collection of raw electronic signal data and live base-calling was performed using the MinKNOW 22.10.10 and the Guppy basecaller (Oxford Nanopore Technologies).

Plasmid content of isolates HUC-90, HUC-99 and HUC-Aba1527 was investigated in detail by hybrid assembly combining the MiSeq short-reads with MinION long-reads using Unicycler v0.5.0.

### 4.6. Plasmid Annotation and Visualization

Circular plasmids were visualized using SnapGene 5.1.4.1 (GSL Biotech, Chicago, IL, USA) and were manually annotated.

### 4.7. Electroporation Experiments

To determine the transferability of *bla*_OXA24/40_, plasmid DNA was isolated from HUC-90, HUC-99 and HUC-Aba1527 using the QIAprep Spin Miniprep Kit (Qiagen, Hilden, Germany) and electroporated into the reference strain *A. baumannii* ATCC 17978. Selection of *A. baumannii* transformants was performed on Luria-Bertani agar (Oxoid, Wesel, Germany) supplemented with ticarcillin (150 mg/L). The presence of *bla*_OXA24/40_ in the obtained transformants was confirmed by PCR.

### 4.8. Conjugation Experiments

The OXA-24/40-encoding plasmids of four isolates, HUC-SM28 (IC2/1999), HUC-90 (IC3/2009), HUC-99 (IC2/2009) and HUC-Aba1527 (IC2/2020), were investigated by broth-mate conjugation experiments to determine the transferability of the *bla*_OXA-24/40_ using the sodium azide-resistant *Escherichia coli* J53 and the rifampicin-resistant *A. baumannii* BM4547 as recipient strains. Selection of *E. coli* J53 transconjugants was performed using sodium azide (200 mg/L), and selection for *A. baumannii* BM4547 was performed using rifampicin (100 mg/L) both combined with ticarcillin (100 mg/L). The transconjugants were tested by PCR for the *bla*_OXA-24/40_ gene and were confirmed by Sanger sequencing the *rpoB* gene.

## 5. Conclusions

Our results illustrate not only the persistence of the IC2 clone over a period of 11 years in the same hospital setting resulting in a monoclonal outbreak in 2020, but also the stability of the plasmid harboring the *bla*_OXA-24/40_ gene in these isolates. Isolates belonging to this clone showed a high degree of multidrug resistance and a variety of virulence factors, what is a threat for patients, and can cause fatal infections.

These findings highlight the difficulty in eradicating *A. baumannii* in hospital settings and the need to strengthen surveillance measures and show that the emergence of *bla*_OXA-24/40_ is of high concern. The presence of environmental isolates belonging to IC2 containing *bla*_OXA-24/40_ gene, emphasizes the need to control the reservoir of resistant isolates to prevent outbreaks.

## Figures and Tables

**Figure 1 ijms-25-02333-f001:**
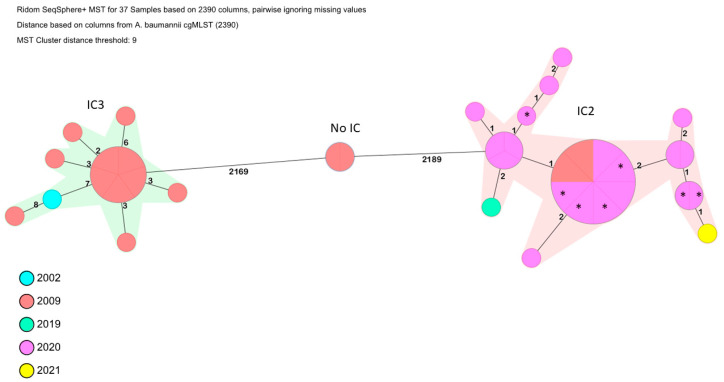
Minimum spanning tree of the CRAb isolates belonging to Outbreak 1 (2009), Outbreak 2 (2020/2021) and control strain HUC-A11 based on 2390 target alleles (*A. baumannii* cgMLST). Numbers between the nodes indicate the number of alleles different. Isolates are coloured based on their year of isolation. Asterisks refer to environmental isolates.

**Figure 2 ijms-25-02333-f002:**
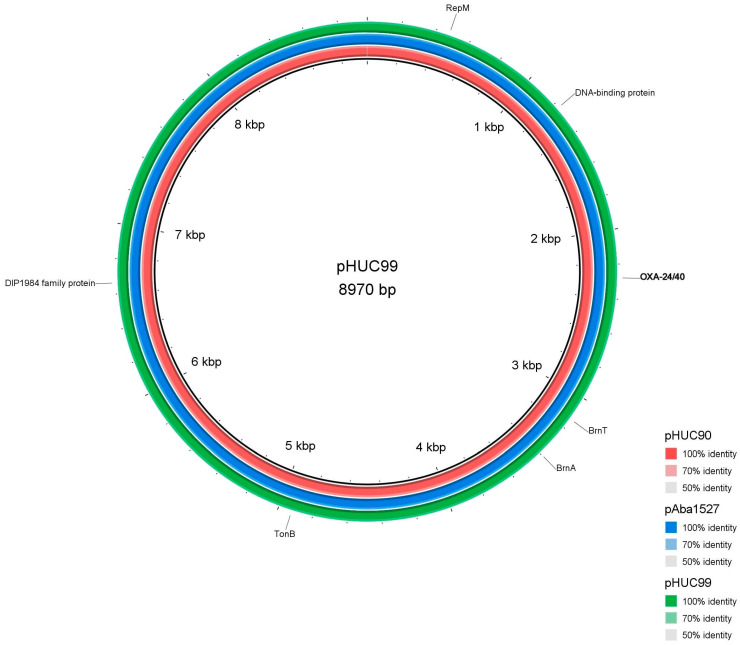
Comparison of the three OXA24/40-encoding plasmids pHUC-90 (red), pAba-1527 (blue), and pHUC-99 (green). The figure was generated using the software BRIG: blast ring image generator.

**Table 1 ijms-25-02333-t001:** Antimicrobial resistance determinants, MLST type, and clonal lineage of the investigated *A. baumannii* isolates. NA: not assigned.

Isolate	Collection Date	MLST		Clonal Lineage	Antibiotic Resistance Determinants					
		STox	STpas		Sulphonamide	Beta-Lactam	Aminoglycoside	Tetracycline	Fluoroquinolones	
									*Gyr*A	*Par*C
HUC-86	18 July 2009	801	2	IC2	*sul1*,*sul2*	*bla*_ADC-30_, *bla*_OXA-66_, *bla*_OXA-24/40_, *bla*_OXA-58_	*aac*(*3*)-Ia-*like*,*aad*A1,*aph*(*3′*)-*VIa-like*,*str*A,*str*B	*tet*(B)-*like*	S81L	S84L
HUC-88	17 February 2009	106	3	IC3	*sul1*	*bla*_ADC-119_, *bla*_OXA-71_, *bla*_OXA-24/40_	*aad*B-*like*,*aph*(*3′*)-*VIa*	-	S81L	S84L
HUC-89	3 June 2009	106	3	IC3	*sul1*	*bla*_ADC-119_, *bla*_OXA-71_, *bla*_OXA-24/40_	*aad*B-*like*,*aph*(*3′*)-*VIa*	-	S81L	S84L
HUC-90	26 June 2009	106	3	IC3	*sul1*	*bla*_ADC-119_, *bla*_OXA-71_, *bla*_OXA-24/40_	*aad*B-*like*,*aph*(*3′*)-*VIa*	-	S81L	S84L
HUC-92	18 September 2009	106	3	IC3	*sul1*	*bla*_ADC-119_, *bla*_OXA-71_, *bla*_OXA-24/40_	*aad*B-*like*,*aph*(*3′*)-*VIa*	-	S81L	S84L
HUC-93	13 August 2009	106	3	IC3	*sul1*	*bla*_ADC-119_, *bla*_OXA-71_, *bla*_OXA-24/40_	*aad*B-*like*,*aph*(*3′*)-*VIa*	-	S81L	S84L
HUC-94	12 August 2009	106	3	IC3	*sul1*	*bla*_ADC-119_, *bla*_OXA-71_, *bla*_OXA-24/40_	*aad*B-*like*,*aph*(*3′*)-*VIa*	-	S81L	S84L
HUC-96	2 July 2009	106	3	IC3	*sul1*	*bla*_ADC-119_, *bla*_OXA-71_, *bla*_OXA-24/40_	*aad*B-*like*,*aph*(*3′*)-*VIa*	-	S81L	S84L
HUC-97	17 June 2009	106	3	IC3	*sul1*	*bla*_ADC-119_, *bla*_OXA-71_, *bla*_OXA-24/40_	*aad*B-*like*,*aph*(*3′*)-*VIa*	-	S81L	S84L
HUC-98	26 June 2009	106	3	IC3	*sul1*	*bla*_ADC-119_, *bla*_OXA-71_, *bla*_OXA-24/40_	*aad*B-*like*,*aph*(*3′*)-*VIa*	-	S81L	S84L
HUC-99	25 June 2009	801	2	IC2	*sul1*,*sul2*	*bla*_ADC-30_, *bla*_OXA-66_, *bla*_OXA-24/40_	*aac*(*3*)-Ia-*like*,*aad*A1,*aph*(*3′*)-*VIa-like*,*str*A,*str*B	*tet*(B)-*like*	S81L	S84L
HUC-100	24 August 2009	106	3	IC3	*sul1*	*bla*_ADC-119_, *bla*_OXA-71_, *bla*_OXA-24/40_	*aad*B-*like*,*aph*(*3′*)-*VIa*	-	S81L	S84L
HUC-101	24 August 2009	106	3	IC3	*sul1*	*bla*_ADC-119_, *bla*_OXA-71_, *bla*_OXA-24/40_	*aad*B-*like*,*aph*(*3′*)-*VIa*	-	S81L	S84L
HUC-106	9 October 2009	128 related	49	NA	-	*bla*_ADC-241_, *bla*_OXA-98_, *bla*_OXA-24/40_	*aad*B-*like*	-	S81L	WT
HUC-109	22 June 2009	128 related	49	NA	-	*bla*_ADC-241_, *bla*_OXA-98_, *bla*_OXA-24/40_	*aad*B-*like*	-	S81L	WT
HUC-A11	25 February 2002	106	3	IC3	*sul1*	*bla*_ADC-119_, *bla*_OXA-71_, *bla*_OXA-1040_	*aad*B-*like*,*aph*(*3′*)-*VIa*	-	S81L	S84L
HUC-SM28	26 April 1999	350	2	IC2	*sul1*	*bla*_ADC-1_, *bla*_OXA-66_, *bla*_OXA-24/40_	*aac*(*3*)-IIa-*like*,*aph*(*3′*)-*VIa*,*str*A,*str*B	*tet*(B)-*like*	S81L	S84L
Hi873	7 December 2019	801	2	IC2	*sul1*,*sul2*	*bla*_ADC-30_, *bla*_OXA-66_, *bla*_OXA-24/40_	*aac*(*3*)-Ia-*like*,*aad*A1,*aph*(*3′*)-*VIa-like*,*str*A,*str*B	*tet*(B)-*like*	S81L	S84L
Aba1516	11 June 2020	801	2	IC2	*sul1*,*sul2*	*bla*_ADC-30_, *bla*_OXA-66_, *bla*_OXA-24/40_	*aac*(*3*)-Ia-*like*,*aad*A1,*aph*(*3′*)-*VIa-like*,*str*A,*str*B	*tet*(B)-*like*	S81L	S84L
Aba1517	11 June 2020	801	2	IC2	*sul1*,*sul2*	*bla*_ADC-30_, *bla*_OXA-66_, *bla*_OXA-24/40_	*aac*(*3*)-Ia-*like*,*aad*A1,*aph*(*3′*)-*VIa-like*,*str*A,*str*B	*tet*(B)-*like*	S81L	S84L
Aba1518	15 June 2020	801	2	IC2	*sul1*,*sul2*	*bla*_ADC-30_, *bla*_OXA-66_, *bla*_OXA-24/40_	*aac*(*3*)-Ia-*like*,*aad*A1,*aph*(*3′*)-*VIa-like*,*str*A,*str*B	*tet*(B)-*like*	S81L	S84L
Aba1519	2 June 2020	801	2	IC2	*sul1*,*sul2*	*bla*_ADC-30_, *bla*_OXA-66_, *bla*_OXA-24/40_	*aac*(*3*)-Ia-*like*,*aad*A1,*aph*(*3′*)-*VIa-like*,*str*A,*str*B	*tet*(B)-*like*	S81L	S84L
Aba1520	15 June 2020	801	2	IC2	*sul1*,*sul2*	*bla*_ADC-30_, *bla*_OXA-66_, *bla*_OXA-24/40_	*aac*(*3*)-Ia-*like*,*aad*A1,*aph*(*3′*)-*VIa-like*,*str*A,*str*B	*tet*(B)-*like*	S81L	S84L
Aba1521	14 July 2020	801	2	IC2	*sul1*,*sul2*	*bla*_ADC-30_, *bla*_OXA-66_, *bla*_OXA-24/40_	*aac*(*3*)-Ia-*like*,*aad*A1,*aph*(*3′*)-*VIa-like*,*str*A,*str*B	*tet*(B)-*like*	S81L	S84L
Aba1522	13 October 2020	801	2	IC2	*sul1*,*sul2*	*bla*_ADC-30_, *bla*_OXA-66_, *bla*_OXA-24/40_	*aac*(*3*)-Ia-*like*,*aad*A1,*aph*(*3′*)-*VIa-like*,*str*A,*str*B	*tet*(B)-*like*	S81L	S84L
Aba1523	13 October 2020	801	2	IC2	*sul1*,*sul2*	*bla*_ADC-30_, *bla*_OXA-66_, *bla*_OXA-24/40_	*aac*(*3*)-Ia-*like*,*aad*A1,*aph*(*3′*)-*VIa-like*,*str*A,*str*B	*tet*(B)-*like*	S81L	S84L
Aba1524	12 October 2020	801	2	IC2	*sul1*,*sul2*	*bla*_ADC-30_, *bla*_OXA-66_, *bla*_OXA-24/40_	*aac*(*3*)-Ia-*like*,*aad*A1,*aph*(*3′*)-*VIa-like*,*str*A,*str*B	*tet*(B)-*like*	S81L	S84L
Aba1525	27 September 2020	801	2	IC2	*sul1*,*sul2*	*bla*_ADC-30_, *bla*_OXA-66_, *bla*_OXA-24/40_	*aac*(*3*)-Ia-*like*,*aad*A1,*aph*(*3′*)-*VIa-like*,*str*A,*str*B	*tet*(B)-*like*	S81L	S84L
Aba1526	20 October 2020	801	2	IC2	*sul1*,*sul2*	*bla*_ADC-30_, *bla*_OXA-66_, *bla*_OXA-24/40_	*aac*(*3*)-Ia-*like*,*aad*A1,*aph*(*3′*)-*VIa-like*,*str*A,*str*B	*tet*(B)-*like*	S81L	S84L
Aba1527	24 October 2020	801	2	IC2	*sul1*,*sul2*	*bla*_ADC-30_, *bla*_OXA-66_, *bla*_OXA-24/40_	*aac*(*3*)-Ia-*like*,*aad*A1,*aph*(*3′*)-*VIa-like*,*str*A,*str*B	*tet*(B)-*like*	S81L	S84L
Aba1528	17 July 2020	801	2	IC2	*sul1*,*sul2*	*bla*_ADC-30_, *bla*_OXA-66_, *bla*_OXA-24/40_	*aac*(*3*)-Ia-*like*,*aad*A1,*aph*(*3′*)-*VIa-like*,*str*A,*str*B	*tet*(B)-*like*	S81L	S84L
Aba1529	16 October 2020	801	2	IC2	*sul1*,*sul2*	*bla*_ADC-30_, *bla*_OXA-66_, *bla*_OXA-24/40_	*aac*(*3*)-Ia-*like*,*aad*A1,*aph*(*3′*)-*VIa-like*,*str*A,*str*B	*tet*(B)-*like*	S81L	S84L
Aba1530	16 October 2020	801	2	IC2	*sul1*,*sul2*	*bla*_ADC-30_, *bla*_OXA-66_, *bla*_OXA-24/40_	*aac*(*3*)-Ia-*like*,*aad*A1,*aph*(*3′*)-*VIa-like*,*str*A,*str*B	*tet*(B)-*like*	S81L	S84L
Aba1531	30 September 2020	801	2	IC2	*sul1*,*sul2*	*bla*_ADC-30_, *bla*_OXA-66_, *bla*_OXA-24/40_	*aac*(*3*)-Ia-*like*,*aad*A1,*aph*(*3′*)-*VIa-like*,*str*A,*str*B	*tet*(B)-*like*	S81L	S84L
Aba1532	24 August 2020	801	2	IC2	*sul1*,*sul2*	*bla*_ADC-30_, *bla*_OXA-66_, *bla*_OXA-24/40_	*aac*(*3*)-Ia-*like*,*aad*A1,*aph*(*3′*)-*VIa-like*,*str*A,*str*B	*tet*(B)-*like*	S81L	S84L
Aba1533	23 October 2020	801	2	IC2	*sul1*,*sul2*	*bla*_ADC-30_, *bla*_OXA-66_, *bla*_OXA-24/40_	*aac*(*3*)-Ia-*like*,*aad*A1,*aph*(*3′*)-*VIa-like*,*str*A,*str*B	*tet*(B)-*like*	S81L	S84L
Aba1534	23 October 2020	801	2	IC2	*sul1*,*sul2*	*bla*_ADC-30_, *bla*_OXA-66_, *bla*_OXA-24/40_	*aac*(*3*)-Ia-*like*,*aad*A1,*aph*(*3′*)-*VIa-like*,*str*A,*str*B	*tet*(B)-*like*	S81L	S84L
Aba1552	4 January 2021	801	2	IC2	*sul1*,*sul2*	*bla*_ADC-30_, *bla*_OXA-66_, *bla*_OXA-24/40_	*aac*(*3*)-Ia-*like*,*aad*A1,*aph*(*3′*)-*VIa-like*,*str*A,*str*B	*tet*(B)-*like*	S81L	S84L

**Table 2 ijms-25-02333-t002:** Virulence factors identified in each International Clone. ATA: Acinetobacter trimeric autotransporter; Type V pili: host cell adhesion; BCRR: Biofilm controlling response regulator; AdeFGH: efflux operon; Csu: chaperone-usher type pilus; BAP: Biofilm associated protein; OMPs: Outer membrane proteins; PNAG: Poly-N-acetyl-D-glucosamine; CPA: Coagulation targeting metallo-endopeptidase of *A. baumannii*. NA: not assigned. * *ata* gene was detected in the 78% of isolates. ** *bfm*S gene was detected in the 82% of isolates. *** *bap* gene was detected in the 36% of isolates.

IC	Adherence and Biofilm Production									Exotoxins	Exoenzymes	Iron Uptake	Capsular Polysacharide	Lipooligosacharide Outer Core
	ATA	Type V Pili	BCRR	*Ade*FGH	Quorum Sensing	Csu Fimbriae	BAP	OMPs	PNAG	Phospholipases C and D	CPA	Acinetobactin		
2	*ata **	*pil* and *fim* genes	*bfm*R, *bfm*S	*ade*F, *ade*G, *ade*H	*aba*I, *aba*R	*csu*C, *csu*D, *csu*E	*bap*	*omp*A	*pga*A, *pga*B, *pga*C, *pga*D	*plc1*, *plc*2, *plc*D	-	*bar*A, *bar*B, *bas*, *and bau*A *genes*	*KL32*	*OCL5*
3	*ata*	*pil* and *fim* genes	*bfm*S ****	*ade*F, *ade*G, *ade*H	*aba*I, *aba*R	-	*bap ****	*omp*A	*pga*A, *pga*B, *pga*C, *pga*D	*plc1*, *plc*2, *plc*D	-	*bar*A, *bar*B, *bas*, *and bau*A *genes*	*KL1*	*OCL1*
NA	*ata*	*pil* and *fim* genes	*bfm*R, *bfm*S	*ade*F, *ade*G, *ade*H	*aba*I, *aba*R	-	-	*omp*A	*pga*A, *pga*B, *pga*C, *pga*D	*plc1*, *plc*2, *plc*D	*cpa*A	*bar*A, *bar*B, *bas*, *and bau*A *genes*	*KL11*	*OCL8*

## Data Availability

The sequence of the *bla*_OXA-1040_ gene is available in GenBank under the accession number OK271078. All the genomes sequences are available under the accession numbers included into the BioProject ID PRJNA1060652 (Appendix A).

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
