# Peer review of "Genomic Surveillance Uncovers a 10-Year Persistence of an OXA-24/40 Acinetobacter baumannii Clone in a Tertiary Hospital in Northern Spain"

_ijms, 2024, doi:10.3390/ijms25042333_

Round 1

Reviewer 1 Report

Comments and Suggestions for Authors

The manuscript presented by Aranzamendi et al. describes two blaOXA24/40 Acinetobacter baumannii outbreaks (2009 and 2020-21) at a tertiary hospital in Northern Spain. The results seem interesting and may be useful from the aspect of monitoring the persistence and surveillance of a virulent IC2 clone in order to control its spread.

Author Response

Dear Reviewer,

We truly appreciate your time and comments to the manuscript ““Genomic surveillance uncovers a 10-year persistence of an OXA24/40 Acinetobacter baumannii clone in a tertiary hospital in Northern Spain”” that have helped to improve it. We have tried to address all your concerns and we are pleased to submit the reviewed manuscript.

Following your recommendations, these are the changes we made in the manuscript:

Major comments:

  1. Page 2, line 100: MLST analysis was performed in order to confirm differences between the strains used in the study. The results are clearly shown but the method is somehow limited, characterizing isolates of the same bacterial species using the sequences of internal fragments of (usually) seven house-keeping genes. In my opinion for this purposes whole genome sequencing is more suitable method due to lower costs, and another benefit of WGS is that no MLST scheme is required. The whole genome sequences of isolates can simply be compared with each other to identify differences, especially in the situation where all strains are isolated from the patients and surfaces in the same hospital. For example, a measure of average nucleotide identity (ANI) can be used to calculate the difference between two strains, and an identity level of 99.9% indicates that the two strains represent the same strain. Due to this reason I still have concerns that the strains belonging to the same IC2 or IC3 and differing in just one or two alleles are actually the same strain persisting in the Hospital Universitario Cruces and the detected differences are spontaneous mutations which are common after multiple cell divisions during time, but not affecting the phenotype of the strain. This is also confirmed by results shown in the table 1 and strains resistotypes provided in the supplementary material (S2 and S3).

In our study we show the core genome MLST results obtained from the short reads assemblies from Illumina (Ridom SeqSphere+ software).To our knowledge the core genome MLST scheme based on the analysis of 2390 genes is the most widely accepted methodology worldwide and a standardized method to compare Acinetobacter baumannii isolates.

As suggested there is a remote possibility that some strains of the same clones differing in a few alleles could be the same one. So that, we have added in the text the corresponding ST following the Oxford scheme where more differences can be seen.

  1. Page 7, line 188: How you can explain the fact that transfer of blaOXA-24/40 by electroporation was successful and conjugation experiments with the control strains were not? In general, my opinion is that chapter Discussion can be significantly improved and strengthened; some results are missing and/or not properly interpreted.

Electroporation experiments are based on the production of an electric discharge onto the recipient bacteria and thus forcedly open some pores in the cell surface to allow the entry of foreing DNA. Conjugation is a natural procedure that depends on the homology of bacterial cells and DNA. To add to this, conjugation requires intrinsic bacterial mechanisms which may be or nor activated in natural conditions. Moreover, especially in A. baumannii, conjugation has been described as a very difficult mechanism to reproduce in vitro.

  1. Page 11, line 352: Similarly, the conclusion seems as repetition of obtained results. The significance and possible impact of the results should be clearly indicated in this chapter.

Done. We have added the corresponding text, see Lines 207-213.

Minor comments:

  1. Page 2, lines 49-50, 52: The names of the bacterial species should be written in italics font

Done

Reviewer 2 Report

Comments and Suggestions for Authors

Please find below my comments on the manuscript:

1.       Introduction, Lines 48-53 “ Acinetobacter baumannii was listed in 2008 as part of the broad ESKAPE group (Enterococcus faecium, Staphylococcus aureus, Klebsiella pneumoniae, A. baumannii, Pseudomonas aeruginosa, and Enterobacter species) [1,2] and it has been recognized as such a big threat, that the World Health Organization (WHO) in 2019 classified carbapenem-resistant A. baumannii (CRAb) as one of the “Priority 1: Critical group” organisms for which new antibiotics are urgently needed [3].” Please write the names of the pathogens in italics.

2.       The type of study in the methods is missing. Please add.

3.       Could the Authors provide information on the susceptibility profile of these isolates to novel antimicrobial agents?

4.       Could the Authors add in Supplementary Table 1 also the outcome (deceased/survived) during the hospitalization of patients included in the study?

5.       Please add the limits of the study in the discussion.

Author Response

Dear Reviewer,

We truly appreciate your time and comments to the manuscript ““Genomic surveillance uncovers a 10-year persistence of an OXA24/40 Acinetobacter baumannii clone in a tertiary hospital in Northern Spain”” that have helped to improve it. We have tried to address all your concerns and we are pleased to submit the reviewed manuscript.

Following your recommendations, these are the changes we made in the manuscript:

  1. Introduction, Lines 48-53 “ Acinetobacter baumannii was listed in 2008 as part of the broad ESKAPE group (Enterococcus faecium, Staphylococcus aureus, Klebsiella pneumoniae, A. baumannii, Pseudomonas aeruginosa, and Enterobacter species) [1,2] and it has been recognized as such a big threat, that the World Health Organization (WHO) in 2019 classified carbapenem-resistant A. baumannii (CRAb) as one of the “Priority 1: Critical group” organisms for which new antibiotics are urgently needed [3].” Please write the names of the pathogens in italics.

Done

  1. The type of study in the methods is missing. Please add.

Done

  1. Could the Authors provide information on the susceptibility profile of these isolates to novel antimicrobial agents?

We thank the reviewer for this interesting idea. However, we do not have novel compounds available to perform the experiments. We will consider this in future projects.

  1. Could the Authors add in Supplementary Table 1 also the outcome (deceased/survived) during the hospitalization of patients included in the study?

Done, all the patients survived and this data is now included as a new column in Table S1.

  1. Please add the limits of the study in the discussion.

Done.

Reviewer 3 Report

Comments and Suggestions for Authors

The article by Maitane Aranzamendi et al., discussing the WGS analyses of 36 Acinetobacter isolates from two outbreaks is very interesting, well written, informative and suitable for publication in this journal. Still very few comments and suggestions are required to be addressed before publication to improve the manuscript and the authors are free to accept or decline the suggestions.

·         Please be sure that all names of bacteria are in italic.

·         The CRAb has been classified by the World Health Organization (WHO) in 2017as one of the “Priority 1: Critical group, not in n2019. Please correct and use the suitable reference.

·         The isolates were assigned to the Pasteur sequence type (ST) 2 (n = 24), ST3 (n = 12) 101 and ST49 (n = 2), totally are 38, while in abstract mentioned 36 isolates, please clarify or correct?

·         Please write the names of genes in Table 1 correctly, for example, tet(B)-like (only tet has to be in italic) strB (the lettes such as B must not be italic), etc.

·         Virulence genes such as adhesion Pil and Fim, have to be italic, please check allover the manuscript and correct.

·         Figure 2 is of low resolution, particularly the keys.

Comments on the Quality of English Language

There are a few english errors need to be address

Author Response

Dear Reviewer,

We truly appreciate your time and comments to the manuscript ““Genomic surveillance uncovers a 10-year persistence of an OXA24/40 Acinetobacter baumannii clone in a tertiary hospital in Northern Spain”” that have helped to improve it. We have tried to address all your concerns and we are pleased to submit the reviewed manuscript.

Following your recommendations, these are the changes we made in the manuscript:

  1. Please be sure that all names of bacteria are in italic.

Done

  1. The CRAb has been classified by the World Health Organization (WHO) in 2017as one of the “Priority 1: Critical group, not in 2019. Please correct and use the suitable reference.

Done

  1. The isolates were assigned to the Pasteur sequence type (ST) 2 (n = 24), ST3 (n = 12) 101 and ST49 (n = 2), totally are 38, while in abstract mentioned 36 isolates, please clarify or correct?

The isolates were 36 although we also analyzed another two control strains (HUC-SM28 and HUC-A11), which it makes a total of 38.

  1. Please write the names of genes in Table 1 correctly, for example, tet(B)-like (only tet has to be in italic) strB (the lettes such as B must not be italic), etc.

Done.

  1. Virulence genes such as adhesion Pil and Fim, have to be italic, please check all over the manuscript and correct.

Done.

  1. Figure 2 is of low resolution, particularly the keys.

Done. Figure 2 was improved and now has a resolution of 300 dpi.